# Novel Three-Dimensional Graphene-Like Networks Loaded with Fe_3_O_4_ Nanoparticles for Efficient Microwave Absorption

**DOI:** 10.3390/nano11061444

**Published:** 2021-05-29

**Authors:** Tao Shang, Qingshan Lu, Jianjun Zhao, Luomeng Chao, Yanli Qin, Ningyu Ren, Yuehou Yun, Guohong Yun

**Affiliations:** 1Department of Physics Science and Technology, Baotou Teacher’s College, Baotou 014030, China; zhaojianjun197174@163.com; 2School of Physical Science and Technology, Inner Mongolia University, Hohhot 010021, China; luqs@imu.edu.cn (Q.L.); qinyli08@126.com (Y.Q.); renningyu217@163.com (N.R.); yunyuehou@163.com (Y.Y.); 3College of Science, Inner Mongolia University of Science and Technology, Baotou 014010, China; 2017993@imust.edu.cn; 4College of Physics and Electronic Information, Inner Mongolia Normal University, Hohhot 010022, China

**Keywords:** graphene-like, networks, Fe_3_O_4_, hybrid, impedance matching, microwave absorption

## Abstract

A novel three-dimensional graphene-like networks material (3D-GLN) exhibiting the hierarchical porous structure was fabricated with a large-scale preparation method by employing an ion exchange resin as a carbon precursor. 3D-GLN was first studied as the effective microwave absorbing material. As indicated from the results of the electromagnetic parameter tests, and the minimum reflection loss (*R*_L_) of the 3D-GLN reached −34.75 dB at the frequency of 11.7 GHz. To enhance the absorption performance of the nonmagnetic 3D-GLN, the magnetic Fe_3_O_4_ nanoparticles were loaded on the surface of the 3D-GLN by using the hydrothermal method to develop the 3D-GLN/Fe_3_O_4_ hybrid. The hybrid exhibited the prominent absorbing properties. Under the matching thickness of 3.0 mm, the minimum *R*_L_ value of hybrid reached −46.8 dB at 11.8 GHz. In addition, under the thickness range of 2.0–5.5 mm, the effective absorption bandwidth (*R*_L_ < 10 dB) was 13.0 GHz, which covered part of the C-band and the entire X-band, as well as the entire Ku-band. The significant microwave absorption could be attributed to the special 3D network structure exhibited by the hybrid and the synergistic effect exerted by the graphene and the Fe_3_O_4_ nanoparticles. As revealed from the results, the 3D-GLN/Fe_3_O_4_ hybrid could be a novel microwave absorber with promising applications.

## 1. Introduction

As the information technology and the electronics industry have been leaping forward, the electromagnetic (EM) wave is being more extensively employed. The use of the EM wave creates convenience in numerous fields, but existing studies suggest that a considerable EM-wave radiation is capable of damaging the immune system, the reproductive system and the nervous system of humans [1]. Moreover, the widespread use of electromagnetic waves can cause EM interference and reduce the stability of information-based communications. Accordingly, the study of microwave-absorbing materials has aroused widespread attention, and the development of a low-density, thin-thickness, wide-bandwidth, and high-absorption absorbing material has always been a research hotspot.

Graphene, as a two-dimensional carbon material, is acting as a potential microwave absorber for its special physical properties of low density, high specific surface area and moderate dielectric loss [2,3]. However, the microwave absorption of conventional graphene is significantly limited [1,4]. The design of the graphene with special structures (e.g., 3D graphene) was able to improve the microwave absorption capacity of graphene [3,5,6]. The self-assembly method and the template-oriented method were employed to prepare the 3D graphene. However, these methods make it difficult to achieve a large-scale preparation, and there are defects (e.g., high costs, complicated processes, and low yields). Moreover, the prepared 3D graphene exhibits poor conductivity and low stability, which adversely affects the effective absorption of microwaves [3,5,6,7,8,9]. Recently, Li et al. developed a facile and large-scale preparation method to fabricate a novel 3D graphene-like material, and the carbon precursor used cheap and easily available ion exchange resin [7,9,10]. As reported from existing studies, the fabricated 3D graphene-like material exhibits a hierarchical porosity, excellent conductivity, and high stability [7,8,9], thereby making it possible to be employed in microwave-absorbing materials.

In addition, the microwave absorption of graphene is only derived from electrical loss, thereby causing a poor impedance matching and a low absorption performance [3,11,12]. To increase the magnetic loss of graphene, a method of hybridizing graphene to magnetic particles is considered effective [11,12]. Among considerable magnetic particle materials, Fe_3_O_4_ has aroused huge attention for its prominent magnetic characteristics, simple preparation and stable chemical properties.

In this study, an ion exchange resin was used as the carbon precursor and the 3D graphene-like networks material (3D-GLN) with hierarchical porosity was prepared with a large-scale preparation method. The microwave-absorption properties of the 3D-GLN were studied initially. More importantly, to increase the magnetic loss, the 3D-GLN/Fe_3_O_4_ hybrid was prepared by using the hydrothermal method initially. As revealed from the results of the tests, the hybrid material exhibited excellent microwave-absorption properties.

## 2. Materials and Methods

### 2.1. Fabrication

The 3D-GLN was prepared by the following steps [7,13,14,15]. The 10 g pretreated macroporous weakly acidic cation-exchange resin was introduced into 100 mL solution of nickelous acetate (0.05 mol/L). After the magnetic stirring for 8 h, the nickel ion exchange resin was washed several times and dried for 12 h (80 °C). Subsequently, 10 g dried nickel ion exchanged resin was impregnated in 500 mL KOH ethanol solution (1 mol/L) and then stirred for 6h. Next, the solution was dried under vacuum at 80 °C, and the product was crushed and then heated in a N_2_ atmosphere at 850 °C for 2 h. Afterwards, the product was sonicated in 3 mol/L HCl solution for 2 h and then rinsed with deionized water until the value of pH reached 7. Lastly, it was dried at 100 °C for 10 h to obtain the black 3D-GLN.

The 3D-GLN/Fe_3_O_4_ hybrid was prepared with a hydrothermal method. To be specific, 0.65 g sodium dodecylbenzenesulfonate (SDBS) was dissolved into 64 mL ethylene glycol (HOC_2_H_4_OH) under magnetic stirring to produce a homogeneous solution. After we added 0.62 g FeCl_3_·6H_2_O and 0.8 g pretreated 3D-GLN, the mixture was stirred for 30 min and then sonicated for 30 min. Next, 1.8 g of NaAc was slowly added to the continuously stirred solution, and then the solution was maintained at 170 °C for 5 h by using a Teflon-lined stainless-steel autoclave. Next, the black product was collected, rinsed several times with ethanol, and then dried at 80 °C. The obtained product was labeled as 3D-GLN/Fe_3_O_4_ (the weight ratio of the Fe_3_O_4_ in the 3D-GLN/Fe_3_O_4_ was 18%). The main chemical reaction equations to form 3D-GLN/Fe_3_O_4_ are expressed below:2Fe^3+^ + HOC_2_H_4_OH → 2Fe^2+^ + CH_3_CHO + 2H^+^(1)
H_2_O + NaAc ↔ NaOH + HAc(2)
5OH^−^ + Fe^2+^ + Fe^3+^ → Fe(OH)_3_ + Fe(OH)_2_(3)
Fe(OH)_2_ + 2Fe(OH)_3_ + 3D-GLN → 3D-GLN/Fe_3_O_4_ + 4H_2_O(4)

Furthermore, the pure Fe_3_O_4_ nanoparticles were prepared for comparison purposes by the above process in the absence of the 3D-GLN.

### 2.2. Characterization

Determination of the phase structure of the product by X-ray diffraction (XRD, 2θ scanning range of 10–80°, Cu Kα radiation, 40 kV and 40 mA, PANalytical Empyrean, Almelo, Netherlands). Raman spectra were obtained using a spectrometer with a 532 nm laser (DXR, Thermo Fisher, Waltham, MA, USA). The morphology and microstructure of the products were observed by a scanning electron microscopic (SEM, Hitachi S-4800, Tokyo, Japan), transmission electron microscopy and high-resolution transmission electron microscopy (TEM and HRTEM, FEI Tecnai F20, Hillsboro, OR, USA). The chemical states of surface elements were investigated by X-ray photoelectron spectroscopy (XPS, Al Kα radiation, Thermo Fisher ESCALAB 250 xi, Waltham, MA, USA). The XPS spectra were calibrated using C 1s signal (284.6 eV). The N_2_ adsorption isotherm measurements were conducted on a Micromeritics ASAP 2020 (Norcross, GA, USA) at −196 °C. The magnetic performances were detected by a physical property measurement system (PPMS, room temperature, Quantum Design Inc., San Diego, CA, USA). We used a vector network analyzer (VNA, frequency range 2–18 GHz, Agilent 8720ET, Santa Clara, CA, USA) to measure EM parameters, including permeability and complex permittivity. For the EM parameters measurements, the sample was mixed with paraffin at a weight ratio of 7: 3. The mixtures were then pressed into a toroidal shape with thickness of 2 mm, outer diameter of 7 mm, and inner diameter of 3 mm.

## 3. Results

### 3.1. The Structure Analysis

The typical powder XRD patterns of the 3D-GLN, the 3D-GLN/Fe_3_O_4_, and the pure Fe_3_O_4_ are presented in Figure 1a. As indicated from the figure, the 3D-GLN showed a significant diffraction peak at 26.2°, belonging to the (002) reflection of graphitic carbon, which demonstrated a high degree of graphitization of the 3D-GLN [15,16,17]. Moreover, the diffraction peak was relatively broad, which suggested that the graphene exhibited a randomly stacked structure; accordingly, the sample was suggested to show a 3D structure [18]. The pure Fe_3_O_4_ displayed six well-defined diffraction peaks that were indexed as the (220), (311), (400), (422), (440), and (511) crystal faces of the face-centered cubic structure of the Fe_3_O_4_, respectively [2,19]. For the 3D-GLN/Fe_3_O_4_ hybrid, the (002) diffraction peak of the graphite carbon and six diffraction peaks of Fe_3_O_4_ were observed. It was therefore indicated that the crystal structure of the 3D-GLN was not destroyed in the process of the hydrothermal synthesis of the 3D-GLN/Fe_3_O_4_, and the face-centered cubic structure Fe_3_O_4_ could be successfully synthesized by using the hydrothermal method in the presence of the 3D-GLN.

Figure 1b presents the Raman spectra of 3D-GLN and 3D-GLN/Fe_3_O_4_. For the tow samples, the peaks at approximately 1355 cm^−1^ (D-band) and 1580 cm^−1^ (G-band) are obvious. D-band belongs to the vibrations of sp^3^ carbon atoms of disorder, as well as the defects in the hexagonal graphitic layers, and G-band belongs to the stretching vibration of sp^2^ carbon atoms within a 2D hexagonal lattice. Thus, the intensity ratio of two peaks (I_D_/I_G_) has been commonly used to estimate the content of defects [20]. Given the data calculation, 3D-GLN and 3D-GLN/Fe_3_O_4_ achieved the I_D_/I_G_ values of 0.38 and 0.65, respectively, which demonstrated that the high temperature reaction of Fe_3_O_4_ loading could cause more defects of 3D-GLN. Moreover, the peak belonging to 2D band is noticeable for the tow samples, which reveals that the pore wall of the material is multilayer grapheme. Furthermore, the peaks at about 540 and 670 cm^−1^ [20] belonging to Fe_3_O_4_ are observed in Raman spectrum of 3D-GLN/Fe_3_O_4_, which further confirms the formation of hybrids.

The morphologies of the 3D-GLN and the 3D-GLN/Fe_3_O_4_ were observed under the SEM (Figure 2a–c). According to Figure 2a, the 3D-GLN and the 3D-GLN/Fe_3_O_4_ exhibited the structure of 3D interconnected porous networks, and their pore sizes were mainly distributed at a sub-micrometer scale. The thin pore walls exhibited a graphene-like structure, and the phenomenon of the crimp could be seen at the outer edges of the walls, which might be attributed to the thin walls composed of a few layers of graphene sheets [7]. The unique structure would lead to the formation of interconnected 3D conductive networks and a large specific surface area, which could enhance the microwave-absorption properties of the samples. Furthermore, considerable Fe_3_O_4_ particles were uniformly loaded on the surface of the pore walls, and the size of the particles was at the nanoscale (Figure 2b,c).

The morphologies and microstructures of the 3D-GLN and the 3D-GLN/Fe_3_O_4_ were further investigated by using the TEM. Figure 3a,b is the TEM images of the 3D-GLN, which show that the sample exhibited the 3D interconnected porous networks and the thin graphene-like pore walls. Moreover, some mesopores were obviously observed on the graphene-like walls. Figure 3b presents that the small mesopores in the range of 2–5 nm were formed by the KOH activation. According to Figure 3c, for the 3D-GLN/Fe_3_O_4_, many nanoscale Fe_3_O_4_ particles were uniformly deposited on the surface of the graphene-like walls, and the diameter of the particles was nearly 10 to 80nm. The Fe_3_O_4_ nanoparticles were attached firmly to the graphene-like walls, even though the ultrasonic treatment was performed during the preparation of the TEM samples, which suggested that a significant adhesion was generated between the Fe_3_O_4_ nanoparticles and the graphene-like walls. Figure 3d presents an HRTEM (high-resolution TEM) photograph of nanoparticles on the graphene-like walls of the 3D-GLN/Fe_3_O_4_. The crystal space was obviously visible, and the interplanar spacing between the lattice fringes was approximately 0.253 nm, corresponding to the (311) crystal planes of Fe_3_O_4_ [12]. This proved that the nanoparticles loaded on the graphene-like walls were face-centered cubic Fe_3_O_4_.

To explore the specific surface areas and porosity properties of the 3D-GLN and the 3D-GLN/Fe_3_O_4_, the N_2_ adsorption/desorption isotherms were measured. According to Figure 4a, the 3D-GLN and the 3D-GLN/Fe_3_O_4_ exhibited the combination characteristics of type II and type IV isotherms, which indicated that they exhibited the multilayer and pore structures. As demonstrated from the BET method analysis, the 3D-GLN and the 3D-GLN/Fe_3_O_4_ exhibited the large specific surface areas of 1135.3 m^2^/g and 857.6 m^2^/g, respectively. Though the specific surface area of the 3D-GLN/Fe_3_O_4_ was narrowed, it remained significantly large. The large specific surface area could contribute to the multiple reflections of EM waves and the enhancement of the electric polarization, thereby probably facilitating the microwave absorption of the samples. Figure 4b plots the corresponding pore size distribution curves derived from the adsorption branches of the isotherms. As indicated from the results, the 3D-GLN and the 3D-GLN/Fe_3_O_4_ exhibited the hierarchical porous structure (e.g., micro-, meso-, and macropores). Compared with the 3D-GLN, the total pore volume of the 3D-GLN/Fe_3_O_4_ decreased, and the pore volume corresponding to the micro- and mesoporous decreased more significantly. This result could be attributed to the fact that the nano-sized Fe_3_O_4_ particles could be easier loaded on the micro- and mesopores, which might cause some pores to be blocked.

The elemental composition and electronic structure of the surfaces of the 3D-GLN and the 3D-GLN/Fe_3_O_4_ were analyzed by using the XPS. Figure 5a presents the XPS survey spectrum of the 3D-GLN and the 3D-GLN/Fe_3_O_4_. According to the figure, both samples showed two peaks at the binding energies of nearly 285 and 531 eV, corresponding to the C1s and O1s peaks, respectively. Moreover, specific to the 3D-GLN/Fe_3_O_4_, a Fe2P peak was identified at about 711 eV, which indicated the presence of Fe elements on the surface of the graphene-like material. The ratio of the C to Fe atom number is 31.5:1 with the sensitivity factor method. Thus, the actual mass ratio of Fe_3_O_4_ in the total material took up 16.96%, insignificantly lower than the theoretical ratio of 18%. This result might be attributed to the adsorption carbon on the sample surface in the XPS test. Figure 5b further presents the high-resolution Fe2p spectrum of the 3D-GLN/Fe_3_O_4_. In the figure, two peaks of the Fe 2p1/2 and the Fe 2p3/2 based on spin orbital splitting were located at 711.3 and 724.5 eV, respectively, which complied with the reported peak positions in the Fe_3_O_4_ [3,11], and there was no appearance of shakeup satellite peak. Thus, the Fe element in the sample could be in the form of Fe_3_O_4_ [21]. The O1s XPS spectrum of the 3D-GLN/Fe_3_O_4_ is illustrated in Figure 5c, and it could be resolved and fitted into three peaks by the Lorentz–Gaussian curve fitting. One of the three peaks was located at around 529.8 eV, belonging to the oxygen in the Fe_3_O_4_ lattice (Fe–O), which further confirmed the presence of the Fe_3_O_4_ nanoparticles [21]. The other two peaks, located at 533.1 and 531.6 eV, corresponding to surface oxygen atoms in the carbonyl oxygen C–O and C=O, respectively, demonstrated the presence of residual oxygen-containing groups (e.g., –COOH and –OH) on the surface of the porous graphene [3,22].

### 3.2. Magnetic Properties

It is generally known that magnetic properties can affect the EM wave-absorbing properties of the material, so the magnetic properties were investigated through magnetization (*M-H*) measurements. Figure 6 presents the magnetization hysteresis loops of the prepared pure Fe_3_O_4_ and 3D-GLN/Fe_3_O_4_ at ambient temperature. Both the magnetization hysteresis loops exhibited an S-like shape, which demonstrated that the Fe_3_O_4_ and the 3D-GLN/Fe_3_O_4_ were the typical ferromagnetic. The saturation magnetization (*M*_s_) value of the prepared pure Fe_3_O_4_ was 54.33 A·m^2^ kg^−1^, lower than that of the reported bulk Fe_3_O_4_ (approximately 92 A·m^2^ kg^−1^) [13]. The lower *M*_s_ could be mainly ascribed to the small size of the pure Fe_3_O_4_ particles prepared with the hydrothermal method [13]. In addition, the 3D-GLN/Fe_3_O_4_ achieved a lower *M*_s_ value of 11.4 A·m^2^ kg^−1^ compared with the pure Fe_3_O_4_ particles, which could be attributed to the existence of the 3D-GLN with no magnetic ordering. Furthermore, the 3D-GLN/Fe_3_O_4_ exhibited a low coercivity and remanent magnetization, which could facilitate the transformation, transmission and attenuation of microwave energy [2].

### 3.3. Microwave Absorbing Properties

The absorption capacity of the absorbers could be characterized by the reflection loss (*R*_L_). In addition, in accordance with the theory of the transmission line, the following equation can be adopted to calculate *R*_L_ (dB) [23]:(5)$$R_{L}=20\;\log\left|\frac{Z_{\mathrm{in}\;}-\;Z_{0}}{Z_{\mathrm{in}\;}+Z_{0}}\right|$$
where *Z*_0_ and *Z*_in_ denote the characteristic impedance of free space and the input impedance of microwave-absorbing materials, respectively. The *Z*_in_ can be given by:(6)$$Z_{\mathrm{in}}=Z_{0}\;{\left(\mu_{r}/\varepsilon_{r}\right)}^{1/2}\;\tan h\left[\;2\pi jdf/c\left(\varepsilon_{r}\mu_{r}\right){\;}^{1/2}\right]$$
where *d* denotes the thickness of the absorbing layer; *c* expresses the velocity of EM waves in free space; *f* represents the frequency of incident EM waves; *μ*_r_ and *ε*_r_ are the complex permeability and complex permittivity of the absorbers measured with the transmission line method. Accordingly, for a microwave-absorbing material, the optimal absorption and impedance matching could be obtained by altering *f* and *d*.

Figure 7 plots the *R*_L_ curves and presents the *R*_L_ contour maps of the 3D-GLN and the 3D-GLN/Fe_3_O_4_ with different thicknesses. According to the figure, the minimum *R*_L_ values of both samples increased first and then decreased with the increase in the thickness. Moreover, as the thickness *d* increased, the minimum *R*_L_ peak shifted obviously to the region of low frequency. The reason could be explained by the following equation based on the quarter-wavelength (*λ*/4) matching model [3,24,25,26]:*t*_m_ = *nc*/4*f*_m_ (|*ε*_r_||*μ*_r_|)^1/2^(7)
where *t*_m_ denotes the thickness of the absorbing material; *f*_m_ expresses the frequency of the incident EM wave. Thus, *f*_m_ was negatively correlated with *t*_m_, thereby indicating that *f*_m_ would decrease with the increase in *t*_m_.

Figure 7a,c shows the absorption characteristics of the 3D-GLN. The minimum *R*_L_ of the 3D-GLN reached −34.75 dB at the frequency of 11.7 GHz with the optimum matching thickness of 3.0 mm. It was significantly stronger than that of the conventional graphene (the minimum *R*_L_ failed to reach −2.5 dB at different thicknesses) [3], the reduced graphene oxide (the minimum *R*_L_ was −6.9 dB under the thickness of 2 mm and the frequency of 7 GHz) [4], and the nitrogen doped graphene (the minimum *R*_L_ was −13.3 dB at nearly 8.2 GHz with the thickness of 4.5 mm) [1]. It is generally known that with *R*_L_ exceeding −10 dB (*R*_L_ < −10 dB), the absorption of microwave energy by the absorber will reach over 90% [27]. Thus, the corresponding frequency range can be termed as the effective absorption bandwidth *f*_e_. Moreover, according to Figure 7a,c, *f*_e_ of the 3D-GLN was 3.3 GHz (10.1–13.4 GHz) under the thickness of 3.0 mm, and it reached 13.1 GHz (4.9–18.0 GHz) with the thickness regulated from 2.0 to 5.5 mm. All the results demonstrated that the prepared 3D-GLN exhibited better microwave-absorption properties than the conventional 2D graphene. More importantly, according to Figure 7b,d, the 3D-GLN/Fe_3_O_4_ exhibited a prominent microwave-absorption performance. First, the minimum *R*_L_ of 3D-GLN/Fe_3_O_4_ was excellent. Under the optimum matching thickness of 3 mm, the minimum RL value of the 3D-GLN/Fe_3_O_4_ reached up to −46.8 dB at 11.8 GHz, which could be better than that of considerable similar absorbing materials reported recently. Second, the *f*_e_ of the 3D-GLN/Fe_3_O_4_ was 4.4 GHz (9.9–14.3 GHz), which could be wider than that of 3D-GLN (3.3 GHz). In addition, such a wide *f*_e_ value was superior over that of the similar materials. Table 1 lists some microwave- absorption properties of the absorbing materials reported recently similar to 3D-GLN/Fe_3_O_4_. Moreover, with the thickness regulated between 2.0 to 5.5 mm, the *f*_e_ reached 13.0 GHz (5.0–18.0 GHz), which involved the entire Ku-band (12–18 GHz), the entire X-band (8–12 GHz), and part of the C-band (4–8 GHz). Accordingly, the 3D-GLN/Fe_3_O_4_ hybrid was suggested as a microwave absorber with promising applications.

To study the absorbing mechanism of the samples, the complex permeability *μ*_r_ = *μ*′ − *jμ*″ and relative complex permittivity *ε*_r_ = *ε*′ − *jε*″ were measured. Figure 8 presents the frequency dependence of *ε*_r_ and *μ*_r_ for Fe_3_O_4_, 3D-GLN and 3D-GLN/Fe_3_O_4_. Figure 8a,b shows that both the values of real part *ε*′ and imaginary permittivity *ε*″ of the 3D-GLN were significantly high. It is generally known that *ε*′ and *ε*″ respectively represent the electric energy storage and dissipation abilities of the materials [27,34,35]. Accordingly, 3D-GLN should exhibit a strong dielectric loss capacity. Ordinarily, the high electric polarization and conductivity of materials can cause high value of *ε*″ [3,36]. The 3D-GLN exhibited the thin graphene-like pore walls, the abundant micro-, meso-, and macropores, and a high specific surface area, which could cause considerable coordination unsaturation atoms and dangling bonds on the surface, thereby endowing the 3D-GLN with a high electric polarization. Moreover, the 3D graphene-like pore walls could form the conductive networks, thereby resulting in the high conductivity of the 3D-GLN. All the mentioned led to the high *ε*″ value of the 3D-GLN. Furthermore, it was reported that with the increase of frequency, *ε*″ of the 3D-GLN decreased. This result could be attributed to the fact that with the increase in the frequency, the space charge polarization decreases and the generation of the displacement current obviously lags behind that of the build-up potential [3,37]. In addition, according to Figure 8a,b, both *ε*′ and *ε*″ of the 3D-GLN/Fe_3_O_4_ were slightly lower than those of the 3D-GLN, which could be attributed to the addition of the Fe_3_O_4_ with a relatively low *ε*′ and a low *ε*″. Thus, both the electric storage capability and dielectric loss of the 3D-GLN/Fe_3_O_4_ could be slightly weaker than those of the 3D-GLN. As shown in Figure 8c,d, compared with ferromagnetic pure Fe_3_O_4_ particles, the 3D-GLN possesses lower *μ*′ and *μ*″ values for its no magnetic ordering. We found that *μ*′ and *μ*″ of the 3D-GLN/Fe_3_O_4_ were higher than those of the 3D-GLN, which could be attributed to the addition of the ferromagnetic pure Fe_3_O_4;_ *μ*′ and *μ*″ respectively represent the material’s ability to store and dissipate magnetic energy [38]. For this reason, the 3D-GLN/Fe_3_O_4_ should exhibit stronger magnetic energy storage and stronger dissipation abilities than the 3D-GLN.

The microwave-absorbing performance of an absorbent was determined by its dielectric loss and magnetic loss. When *ε*_r_ and *μ*_r_ met the impedance matching conditions, the tan*δ*_e_ = *ε*″/*ε*′ (dielectric loss tangent) and tan*δ*_m_ = *μ*″/*μ*′ (magnetic loss tangent) could be used to evaluate the dielectric loss and the magnetic loss, respectively [39]. The calculated tan*δ*_e_ and tan*δ*_m_ of the Fe_3_O_4_, the 3D-GLN and the 3D-GLN/Fe_3_O_4_ are presented in Figure 9. The results show that among the three samples, the 3D-GLN exhibited the highest tan*δ*_e_ and the lowest tan*δ*_m_, and its tan*δ*_e_ was significantly higher than tan*δ*_m_. Moreover, a peak of tan*δ*_e_ was identified at the frequency of 11.7 GHz, which was identical to the frequency of the minimum *R*_L_. Accordingly, the dielectric loss could primarily account its high microwave absorption. Compared with the 3D-GLN, the tan*δ*_e_ of the 3D-GLN/Fe_3_O_4_ slightly decreased, whereas tan*δ*_m_ greatly increased. This result demonstrated that the improvement of microwave absorption of the 3D-GLN/Fe_3_O_4_ was primarily determined by the increase in the magnetic loss.

According to the theory of the transmission line, a high-quality EM absorbent should be met with a good impedance matching, i.e., more EM waves can be incident into the absorber rather than being reflected. The impedance matching of materials could be characterized by the value of normalized input wave impedance Z, as illustrated by the following equations [40]:(8)$$Z=\left|Z_{\mathrm{in}}/Z_{0}\right|$$
where *Z*_in_ can be calculated by Equation (6). And when the ideal optimal impedance matching condition was reached, the Z value was equal to 1 (*Z*_in_ = *Z*_0_). The *Z-f* curves of the 3D-GLN and 3D-GLN/Fe_3_O_4_ with a thickness of 3 mm are plotted in Figure 10a. According to the figure, the *Z* values of the 3D-GLN and the 3D-GLN/Fe_3_O_4_ showed the broad peaks around 11.7 GHz, which basically complied with the frequency of their minimum *R*_L_ peeks. Moreover, the *Z* value of the 3D-GLN/Fe_3_O_4_ increased compared with that of the 3D-GLN, especially in the high frequency range, thereby indicating that the 3D-GLN/Fe_3_O_4_ achieved the better impedance matching. This result could be attributed to the addition of Fe_3_O_4_, which could adjust the EM parameters of the hybrid. Moreover, the Fe_3_O_4_ nanoparticles growing on the surface of graphene can act as the antenna receivers, so EM waves penetrated maximally into the interior of the absorber [41].

A good impedance matching could enable considerable EM waves to enter the absorber. Moreover, specific to an excellent microwave-absorbing material, the absorption inside the absorber is of a high significance. The internal absorption capacity could be evaluated by the attenuation constant α which is determined below [27]:(9)$$\alpha =\frac{\sqrt{2}\pi f}{c}\times \sqrt{\left(\varepsilon^{\prime \prime }\mu^{\prime \prime }-\mu^{\prime }\varepsilon^{\prime }\right)+\sqrt{{\left(\varepsilon^{\prime }\mu^{\prime \prime }+\mu^{\prime }\varepsilon^{\prime \prime }\right)}^{2}+{\left(\varepsilon^{\prime \prime }\mu^{\prime \prime }-\varepsilon^{\prime }\mu^{\prime }\right)}^{2}}}$$

The frequency dependence of α for the 3D-GLN and the 3D-GLN/Fe_3_O_4_ are calculated and shown in Figure 10b. It can be seen that in the test frequency range, *α* value of the 3D-GLN/Fe_3_O_4_ was higher than that of the 3D-GLN. Moreover, the enhancement of *α* value could be ascribed to the addition of the Fe_3_O_4_, which could increase the magnetic loss capability of the sample.

In addition, the special structure morphologies of the 3D-GLN and the 3D-GLN/Fe_3_O_4_ could largely account for their excellent microwave-absorbing performance. First, the 3D-GLN and the 3D-GLN/Fe_3_O_4_ exhibited the 3D interconnected conductive networks composed of the graphene-like pore walls, which also had certain resistance. When EM waves were incident, the resistance–inductance–capacitance coupling circuits and the time-varying EM induction currents could be formed in the 3D networks. Such long-distance induced currents were rapidly attenuated in the resistor networks and transformed into thermal energy, thereby causing incident EM waves to be attenuated rapidly. In addition, compared with graphene, 3D-GLN has more defects and functional groups. The defects and groups can not only improve the impedance match characteristic but also introduce the transition from contiguous states to Fermi level, defect polarization relaxation, and groups’ electronic dipole polarization relaxation, which could all contribute to electromagnetic wave penetration and absorption. Moreover, the 3D-GLN and the 3D-GLN/Fe_3_O_4_ exhibited abundant micro-, meso-, and macropores. Thus, the multiple reflection phenomena of EM waves would occur in the channels and cavities of the samples, which could increase the propagation distance and absorption region of EM waves in the absorbers. Second, for the 3D-GLN/Fe_3_O_4_, the Fe_3_O_4_ nanoparticles improved the impedance matching and the magnetic loss of the sample, while creating considerable EM wave scattering points, which led to the increase in the propagation distance and absorption region of EM waves. Third, the destructive interference could occur when the incident, reflected and scattering EM waves propagated in the channels and cavities of the 3D networks; as a result, the energy of EM waves would be reduced effectively. Lastly, the introduction of Fe_3_O_4_ particles could induce considerable interfacial polarization losses, which originated from the interfaces between the Fe_3_O_4_ particles and between the Fe_3_O_4_ particles and graphene-like walls. The mentioned discussions can be illustrated by the schematic diagram Figure 11.

## 4. Conclusions

In brief, a novel low-density carbon material 3D-GLN was prepared with a large-scale preparation method by employing the metal ion-exchange resin as a precursor. The 3D-GLN exhibited the interconnected conductive networks and the hierarchical pore structure composed of the micro-, meso-, and macropores. As indicated from the measurement of the EM parameters, the 3D-GLN exhibited a high dielectric loss and the good microwave-absorbing properties. The minimum *R*_L_ of the 3D-GLN reached −34.75 dB at the frequency of 11.7 GHz with the optimum matching thickness of 3.0 mm. To further enhance the absorption performance of the 3D-GLN with no magnetic ordering, the Fe_3_O_4_ nanoparticles were deposited on the surface of the 3D-GLN via the hydrothermal process, and a prominent microwave-absorption performance was obtained. The minimum *R*_L_ value of the 3D-GLN/Fe_3_O_4_ hybrid reached up to −46.8 dB at 11.8 GHz, and *f*_e_ reached 4.4 GHz. By adjusting the thickness between 2.0 to 5.5 mm, *f*_e_ could also reach 13.0 GHz, which covered part of the C-band and the whole X-band and Ku-band. The remarkable microwave absorbing could be explained by the special 3D structure and the effect of loading Fe_3_O_4_.

## Figures and Tables

**Figure 1 nanomaterials-11-01444-f001:**
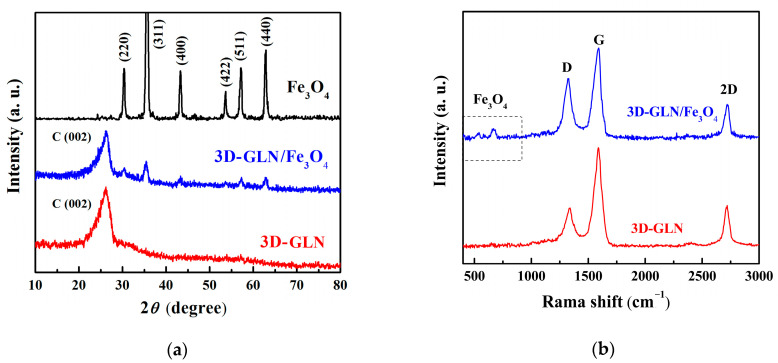
(**a**) XRD patterns of 3D-GLN, 3D-GLN/Fe_3_O_4_, and pure Fe_3_O_4_, (**b**) Raman spectra of 3D-GLN and 3D-GLN/Fe_3_O_4_.

**Figure 2 nanomaterials-11-01444-f002:**
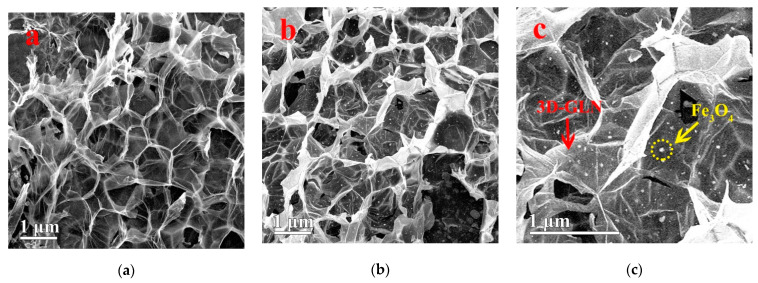
The SEM images of 3D-GLN (**a**) and 3D-GLN/Fe_3_O_4_ (**b**,**c**).

**Figure 3 nanomaterials-11-01444-f003:**
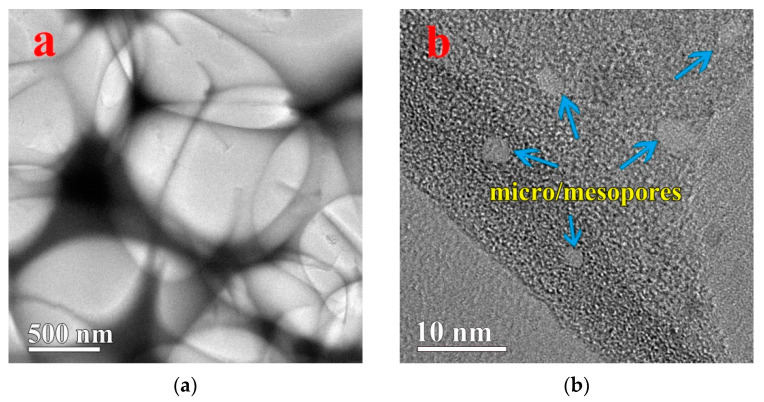
The TEM images of 3D-GLN (**a**,**b**) and 3D-GLN/Fe_3_O_4_ (**c**), and the HRTEM images of 3D-GLN/Fe_3_O_4_ (**d**).

**Figure 4 nanomaterials-11-01444-f004:**
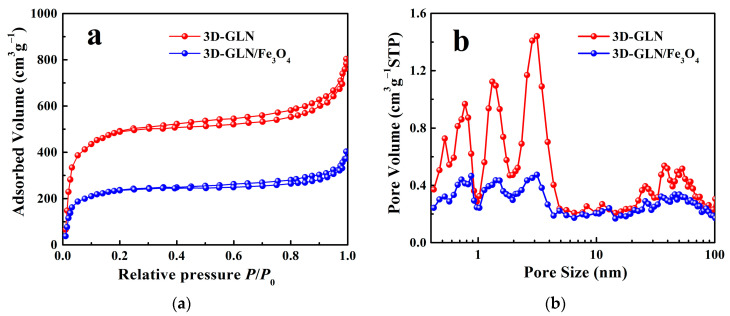
Nitrogen adsorption/desorption isotherms (**a**), the pore size distributions of 3D-GLN and 3D-GLN/Fe_3_O_4_ (**b**).

**Figure 5 nanomaterials-11-01444-f005:**
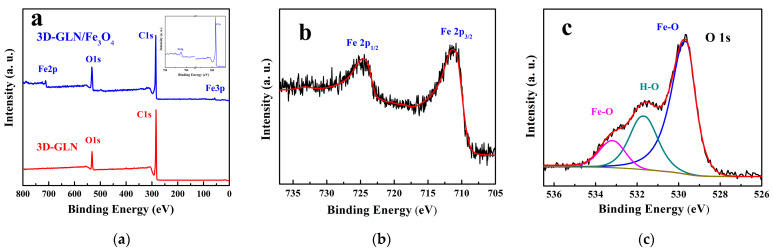
XPS survey spectra of 3D-GLN and 3D-GLN/Fe_3_O_4_ (**a**). The Fe2p (**b**), and O1s (**c**) XPS spectra of 3D-GLN/Fe_3_O_4_.

**Figure 6 nanomaterials-11-01444-f006:**
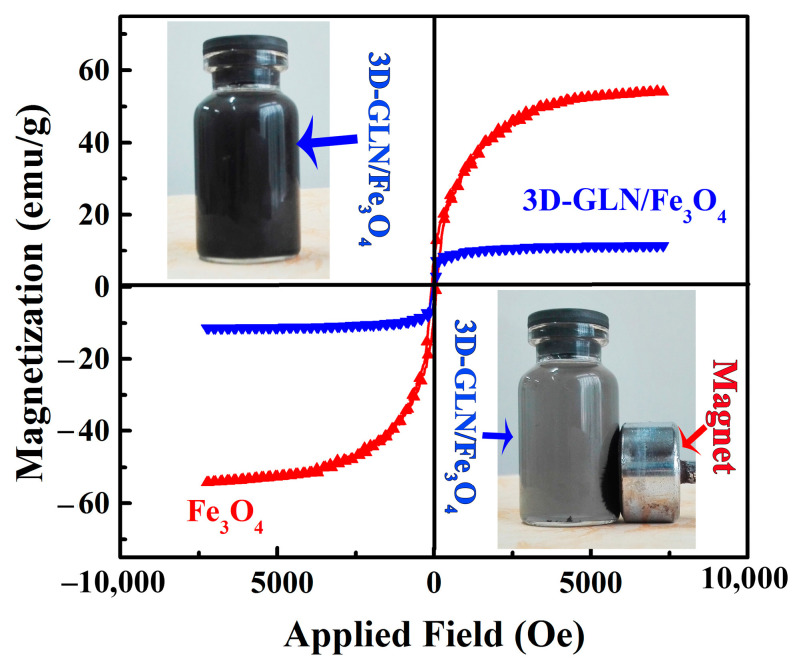
Magnetization curves of pure Fe_3_O_4_ and 3D-GLN/Fe_3_O_4_ at ambient temperature.

**Figure 7 nanomaterials-11-01444-f007:**
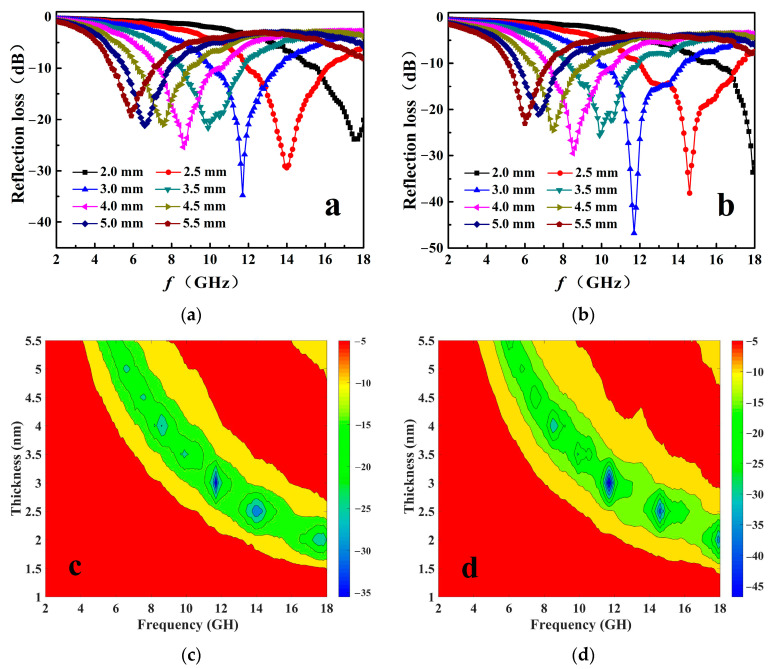
The reflection loss (*R*_L_) of 3D-GLN (**a**) and 3D-GLN/Fe_3_O_4_ (**b**) with different thicknesses. The *R*_L_ contour maps with different thickness and frequency of 3D-GLN (**c**) and 3D-GLN/Fe_3_O_4_ (**d**).

**Figure 8 nanomaterials-11-01444-f008:**
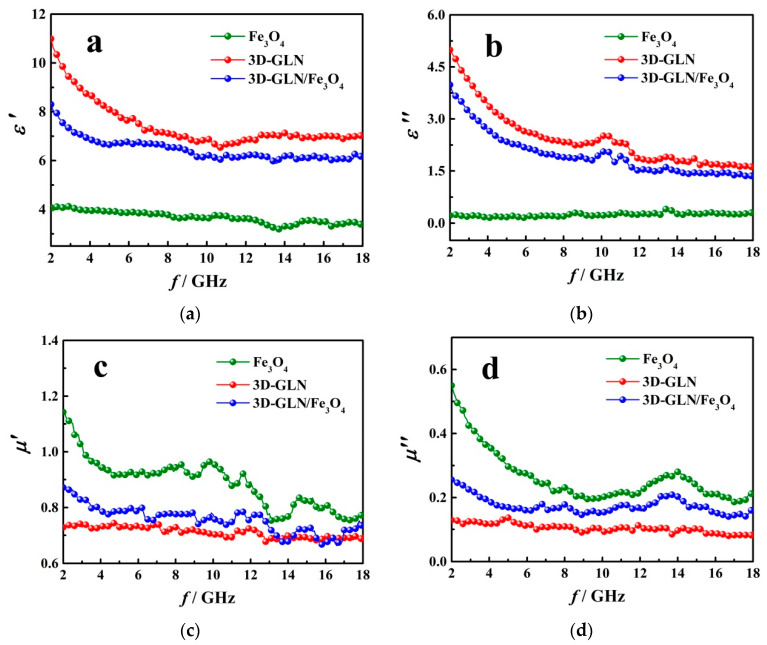
Frequency dependence of *ε*′ and *ε*″ (**a**,**b**) and *μ*′ and *μ*″ (**c**,**d**) for Fe_3_O_4_, 3D-GLN and 3D-GLN/Fe_3_O_4_.

**Figure 9 nanomaterials-11-01444-f009:**
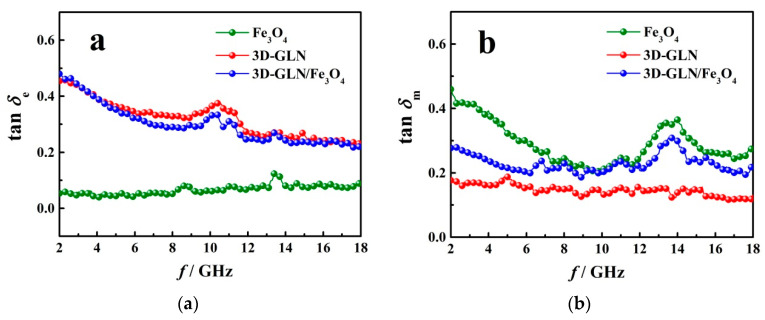
Frequency dependence of tan*δ*_e_ (**a**) and tan*δ*_m_ (**b**) of Fe_3_O_4_, 3D-GLN and 3D-GLN/Fe_3_O_4_.

**Figure 10 nanomaterials-11-01444-f010:**
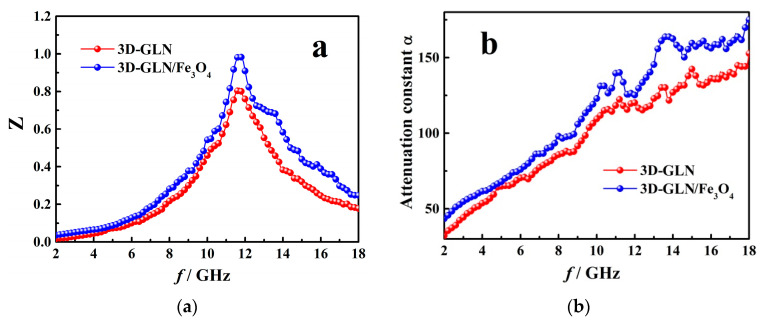
Frequency dependence of *Z* (**a**) and *α* (**b**) for the 3D-GLN and the 3D-GLN/Fe_3_O_4_.

**Figure 11 nanomaterials-11-01444-f011:**
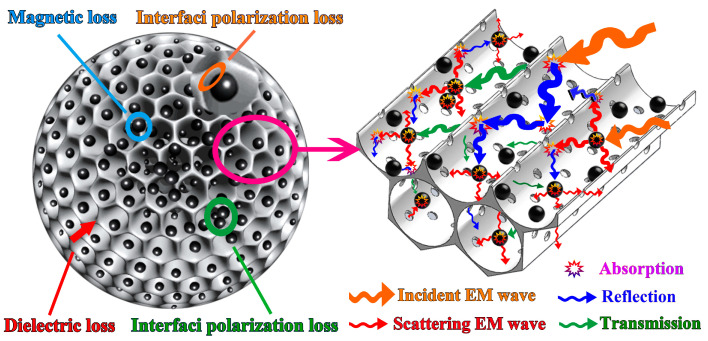
The schematic diagram of the possible microwave loss mechanism for the 3D-GLM/Fe_3_O_4_.

**Table 1 nanomaterials-11-01444-t001:** Microwave-absorption performances of similar absorbing materials reported recently.

Materials	Minimum *R*_L_	Effective Absorption Bandwidth *f*_e_	Reference
3D-GLN	−34.75 dB	3.3 GHz	This work
3D-GLN/Fe_3_O_4_	−46.8 dB	4.4 GHz	This work
3D graphene–Fe_3_O_4_ nanocomposites	−23.0 dB	4.8 GHz	[6]
3D graphene@Fe_3_O_4_@Ppy	−40.53 dB	5.12 GHz	[28]
3D interlinked ACNT/RGO/BaFe_12_O_19_	−19.03 dB	3.8 GHz	[29]
Fe_3_O_4_/rGO	−22.7 dB	3.13 GHz	[2]
rGO-Fe_3_O_4_ nanocomposites	−37.5 dB	1.9 GHz	[30]
Hollow Fe_3_O_4_@ RGO	−41.89 dB	4.2 GHz	[31]
Graphene@Fe_3_O_4_@PANI@WO_3_	−46.8 dB	1.8 GHz	[32]
Fe_3_O_4_/SiO_2_/graphene	−27.1 dB	2.5 GHz	[33]

## Data Availability

The data presented in this study are available on request from the corresponding author.

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
