# Peer review of "Novel Three-Dimensional Graphene-Like Networks Loaded with Fe3O4 Nanoparticles for Efficient Microwave Absorption"

_nanomaterials, 2021, doi:10.3390/nano11061444_

Round 1

Reviewer 1 Report

The authors have prepared a novel three-dimensional graphene-like networks loaded with Fe3O4 nanoparticles. This material is presented as an efficient microwave absorber.

The material has been adequately characterized by X-ray diffraction, as well as other methods.

Obviously, this novel material has excellent properties, superior to other similar materials described previously. However, in my opinion the authors should be more specific about the potential applications of this new three-dimensional graphene-like network as a microwave absorber. If the authors think that this material can be useful as a microwave absorber in chemical synthesis, they should detail its properties at 2.45 GHz. They should also detail potential applications as a function of thickness at 11 GHz.

Reviewer 2 Report

This paper reports fabrication of composite material of graphene-like network structure decorated with Fe3O4 nanoparticles for microwave absorption material. 3D-GLN was fabricated from ion exchange resin carbon precursor. The addition of Fe3O4 to 3D-GLN was effective in improving the microwave absorption property. Here are the comments to the authors.

  • Please revise the chemical formula (1). The EG is not on the right side.
  • What is a “graphene-like” network? Is this graphene or carbon sheet? What is the exact structure of carbon (sp2 rich or sp3 rich)?
  • How about the stability of the 3D-GLN structure after deposition of Fe3O4 at the relatively high-temperature reaction. Maybe Raman spectroscopy is effective in evaluating the possible damage of the treatment to the 3D-GLN. In the Nitrogen absorption test, total pore volume for 3D-GLN/Fe3O4 was largely decreased from 3D-GLN. This can be because of the deposition of Fe3O4 but also a deconstruction of 3D-network structure.
  • How the amount of Fe3O4 was determined? Is this the optimized amount to exhibit good microwave absorption?
  • How big can the 3D-GLN/Fe3O4 sheet be fabricated?
  • This material has an improved property at C and X-band while the lower frequency was not tested. The penetration depth of microwaves to materials is wider at a lower frequency. Maybe it is necessary to cover lower frequency.
  • What is the effect of GLN structure for exhibiting microwave shielding property? Should it be a graphene-like structure or network structure? Is there any advantage as compared to other carbon materials?
  • What is the advantage of present materials as compared to similar EM shielding materials? The author should highlight the excellent performance compared to existing materials.

Round 2

Reviewer 1 Report

The authors have satisfactorily resolved in the revised version of the manuscript all previously raised issues.
I therefore recommend the publication of the manuscript in its current state.

Reviewer 2 Report

The authors revised the paper accordingly to the comments. I believe that this version can be accepted in Nanomaterials.